# Trajectories of Resilience in University Inductees following Outdoor Adventure (OA) Residential Programmes

John F. Allan * and Jim McKenna

Carnegie School of Sport, Leeds Beckett University, Leeds LS16 5LF, UK; j.mckenna@leedsbeckett.ac.uk
* Correspondence: j.allan@leedsbeckett.ac.uk; Tel.: +44-785-230-2124

**Abstract:** Even before COVID-19, university inductees were vulnerable to transitional stressors, which impact upon their psychological well-being and ability to complete their studies. Resilience, as a psychological construct, may be analogous to holding the functioning that supports higher-level academic performance in twenty-first century higher education (HE). Given the significant investment HE requires, coupled with students' capacity to develop their resilience, universities may be expected to promote psychological resilience in new students. Linking HE to resilience, outdoor adventure (OA) residential programmes have enabled inductees to build components of resilience (i.e., increased self-perception, feelings of control, and intrapersonal relationships) that may heighten their immediate and longer-term academic performance. Yet, few studies have examined the sustainability of these effects. Across five annual cohorts, the self-perceived resilience of 2500 inductees was profiled across three time-point trajectories: (i) pre-OA programme, (ii) post-OA programme, and (iii) three months post-OA programme. Further, the functionality of inductees' enduring resilience was evaluated for predicting their prospective academic performance at the end of their first year of study. Students' self-perceived resilience, well-being, and positive recollection of OA experiences reflected their bounce-back ability and a healthy trajectory of productive functioning. Students reporting higher levels of resilience after three months of following the programme were more likely to achieve better prospective academic outcomes. The large sample size ensured that a powerful detection of change was established across time. However, given the absence of a comparison condition across all time points, any lasting improvements in students' resilience was unable to be attributed to the OA programme. Nonetheless, the results give significant grounds for further research in this direction, including the study of more distinct narrative enquiries at follow-up. In this way, pedagogical practices, supporting effective interventions, may be deployed with incoming students, with the aim of building and maintaining students' on-going resilience across different learning contexts.

**Keywords:** university inductees; psychological resilience; academic performance; outdoor adventure intervention programmes; robust follow-up empirical design; adaptable productive functioning



## 1. Introduction

The global reach of the coronavirus disease 2019 (COVID-19) has presented significant challenges for the psychological well-being and adaptive capabilities of young people [1]. Individuals under 25 years of age have reported worsening mental health and ability to cope during the pandemic than their older counterparts, aged over 65 [2]. Even before the effects of COVID-19 and lockdowns emerged, there were enduring concerns about the psychological development of the current cohort of young people. Risks to their mental health and productive functioning may come in the form of unemployment, over-reductive education, fast-changing technological advancements, high divorce rates, media intrusion, and consumerism [3–7]. The incapacity to cope with an accumulation of stressors will have caused some students entering universities to present low adaptive capabilities and heightened psychological distress. For young people, the transition into university represents

a pivotal 'in between' period, requiring adjustment to the interim between dependence on the family and complete independence [8]. Starting university will require students to manage their academic progress alongside a possible influx of stressors, including financial uncertainty, time constraints, establishing new relationships and conforming to new social norms [9,10]. Persistent exposure to combinations of stressors in an unfamiliar environment and without established social networks make higher education (HE) inductees particularly vulnerable to psychological and physical health problems, leading, in some cases, to early drop-out [11–13]. Mental ill-health and low adaptive functioning within new university student populations has been reported to be problematically high [14–16] and evidenced in multiple countries [17–20]. This makes student mental health an important contemporary public health issue.

Psychological resilience constitutes a range of positive adaptive behaviours, which may enable new students to combat potential mental health problems and avoid a premature departure from university [21]. While resilience does not ensure mental health, these behaviours help individuals to solve problems, deal with setbacks, manage work conscientiously, communicate with people from a variety of backgrounds, and adapt rapidly to changing conditions [22]. Resilience has protected university students against negative responses to risk [23], improved their mental health during stressful exam periods [24], and may more strongly influence retention and achievement in HE than qualifying grades [25]. Therefore, highly resourceful individuals with functional cognitive, social, and emotional repertoires for handling the demands of HE may be considered resilient.

Under the premise of optimising student integration into a new environment [26–29], one-week residential outdoor adventure (OA) programmes have reported immediate and lasting increases in students' resilience, compared to non-attendees. Multivariate statistical analyses, evaluating the immediate impact of the one-week OA programming in the present study, reported an 8.35% positive gain in over 2500 students' resilience, compared to non-attendees. Camp-based experiences, such as mastering new skills, developing new relationships, and being female, predicted their heightened resilience [30]. From the few studies evaluating the lasting impact of OA programming on students' resilience [31–34], findings support the notion that outdoor environments provide dynamic contexts and experiences for building repertoires of successful adaptive functioning. These include perseverance, social support, and a sense of belonging. Even allowing for this emerging understanding, greater clarity is still required, regarding the sustainability of these changes and the use of resilience as a trajectory of positive adaptation for new students.

### 1.1. Resilience as a Model Trajectory in Higher Education (HE)

Resilience is a diffuse concept derived from an inter-play of 'individual-environmental' transactions [35–37]. It also relates to the acquisition of powerful promotive and protective developmental competencies for handling future difficulties [38]. These factors appear at the individual level, such as self-regulation, optimism, internal locus of control and self-esteem within family, and secure attachments (e.g., sociability and empathy), as well as through broader social and community values (e.g., education) [39]. Although there may be significant differences in how young people respond to disadvantage and risk [40], demonstrating competent functioning across difficult circumstances and domains provides evidence of the enduring functionality of resilience [41]. Given psychosocial problems, such as low self-esteem, substance abuse, and school drop-out, are among the most common chronic health disorders of youth, and positive adaptive functioning is acknowledged as a vital sign of life [42].

Yet, confusion remains regarding if/how resilience provides sustainable protective resources for young people in 'real world' scenarios. For example, resilience factors that promote positive psychosocial functioning in one context, a time-point, or a cohort may be ineffective in another [43,44]. Moreover, resilience and psychological well-being are distinctive [45]. Therefore, emotional pain and resilience may co-exist on the road to positive adaptation.

Resilience makes up one of many trajectories into mental health, including recovery, which may be manifest following successful responses to a broad range of losses, uncertainties, or challenges. Figure 1 highlights three hypothesised trajectories of responses to stress, determined from longitudinal research on trauma and bereavement [46].

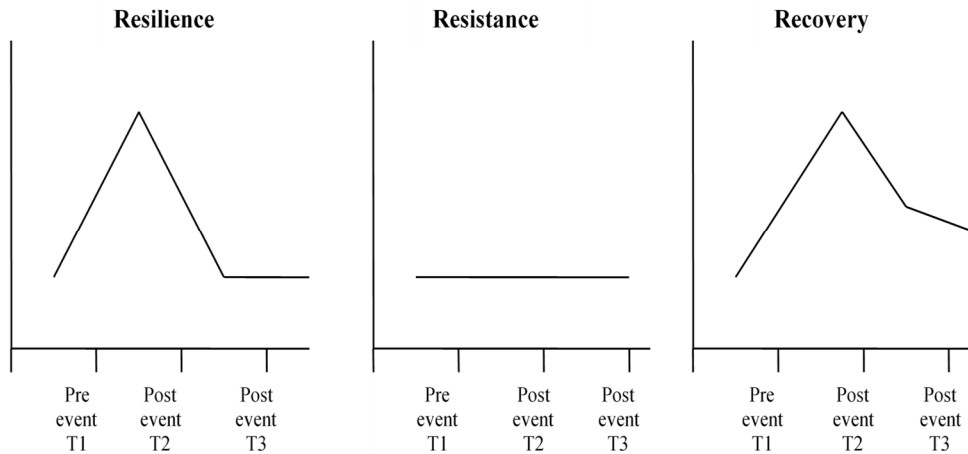

**Figure 1.** Hypothesised trajectories of the course of stress response.

Figure 1 highlights that resilience represents a reasonably rapid return to homeostasis once displaced (pre-event T1 to post-event T3), suggesting 'bounce-back-ability' and a stable trajectory of healthy functioning [47,48]. This transient perturbation, shown as a 'witches hat', may last between weeks and months [49]. Resilience is contrasted from the other trajectories by a temporally different response to adversity. Resistance minimises dysfunction because coping resources are present at the time of displacement (T2); therefore, post-event realignment at T3 is unlikely. Recovery involves a prolonged period of dysfunction (a threshold psychopathology, e.g., depression) lasting several months or more (T2), followed by a gradual return to pre-event functioning. Although stress responses are highly individualised, this set of trajectories helps to explain the patterns of behaviours following exposure to trauma.

To draw further inferences, regarding the consequences of stress and resilience over time, research needs to be conducted to analyse post event waves in direct comparison to pre-event measures [50]. Theoretically, resilience comprises of a common set of healthy behaviours that enable individuals to return to a state of equilibrium (surviving), in order to developing conditions of flourishing or thriving [51]. This suggests resilience may also be analogous to generative experiences and personal growth. To understand resilience as a distinct trajectory of positive adaptation, clarity is required, regarding the nature of resilient responses across time within and across groups. This includes the speed by which homeostasis is re-established, or otherwise, as well as the appropriate blend of promotive and protective factors (subdomains), which may underpin or promote sustainable resilience.

Given that most students who drop-out from, or fail, HE courses do so in their first year [52,53], staff in higher education institutions (HEIs) need to understand the nature of lasting resilience, to help different cohorts of inductees navigate the fluctuating demands of university life. The notion of *academic resilience* or *academic buoyancy* [54] has been proffered to reflect a student's ability to successfully deal with educational setbacks, stress, and study pressures in schools (e.g., [55]). University students with high resilience report less psychological distress and elevated levels of perceived social support, campus connectedness, and academic achievement than their less resilient peers [56–59]. A relational form of resilience, adopted by female undergraduate students, has enabled them to deal with stress and is more facilitative for their subsequent academic performance than the resilience reported by male students [17,60,61]. Despite these findings, HE is widely associated with inadequate pedagogy and support strategies for fully assimilating students. This has led to calls for universities to build the resilience of students. In this way, they can overcome a tide of

increasing mental illness and create stronger adaptive pathways to success within and beyond the first year of study [62].

### 1.2. Outdoor Adventure Residential (OA) Programming for Building Lasting Resilience

Despite substantial advocacy, there continues to be a dearth of evidence to support using OA for the psycho-social development of young people. All five meta-analyses of previous research [63–67] conclude that, although there is variability in programme design and extent of outcome, OA impacts positively on young people's attitudes, emotional well-being, beliefs, and self-perceptions, as well as their social and cognitive capabilities. Importantly, this is not just the case for able and motivated youngsters; under-achievers also perform better in a natural environment, especially when exposed to high-quality, stimulating activities.

At face value, there is a 'fit' between the stated goals of OA residential programming and needs of a wide range of students. Exposing incoming students to OA-based challenges means subscribing to principles adopted by adolescent models of resilience; these range from compensatory, protective, challenging, and inoculation models. Fundamentally, they entail experiencing low–moderate levels of risk evoking mild disruptions in normal functioning to build situational and on-going protective resistance. This is proposed to have a 'steeling effect' for later stress [43,68]. OA programmes, specifically designed to help US university inductees, have demonstrated lasting improvements in their autonomy, interpersonal relationships, and future academic achievement, over comparable non-attendees [69,70].

It is surprisingly then that few studies have specifically evaluated the continual impact of OA programming on new students. Across a continuum of resilience interventions for 'at risk' students, delivered in wilderness settings, qualitative data concluded that relational elements emanating from this exposure reflected the core components of good clinical interventions [71]. An anti-bullying OA intervention programme, assessing the resilience assets of 51 students, indicated that the programme positively affected resilience traits, such as self-efficacy, across three time points [72]. Female students demonstrated greater gains than males. Their study concluded that future research should assess how levels of resilience impacted educational outcomes.

In a three-week OA progrmame, university inductees developed six narrative themes for developing their resilience: perseverance, social support, confidence, responsibility to others, self-awareness, and achievement. [32,73]. It was concluded these meaningful experiences generated enduring capabilities that could be transferred to other contexts. The sustainability of resilience of 72 new students was reported, following a 10-day developmental sea voyage [31]. As there was no change in resilience scores from the last day of the voyage to five months later, it was concluded that resilience was maintained. In a follow-up study, 60 students increased their resilience because of a perceived sense of belonging nine months, following a similar sea voyage experience [34].

This research offers promise for helping inductees into and through HE. However, substantial methodological limitations have prevented widespread adoption of OA experiences for this purpose. Attributing cause and effect has been problematic; effects may be linked to exposure to specific experiences [74,75]. It is also possible that any developmental outcomes may be situation-specific and fail to transfer to everyday settings [76]. Further limitations of current studies include small sample sizes, over-reliance on anecdotal evidence, poor instrumentation, weak theorising, and inadequate empirical evidence to link resilience, the mediating effects of gender, and educational achievement.

These lingering uncertainties present problems for justifying the use of OA programming to help students make effective transitions into university. It also makes it difficult to move beyond simple questions of exposure, as well as the more complex and realistic questions of 'if and when' top-up experiences may be required to sustain beneficial outcomes. Establishing the links between sustained resilience and inductees' prospective academic outcomes may also help to confirm or dispute the important role that resilience plays in

supporting future success in HE. The present study sought to investigate the sustainability and functionality of university inductees' psychological resilience, following a five-day OA residential programme. Empirical evidence was systematically evaluated from a large sample of students within and between five consecutive annual cohorts.

### 1.3. Research Aims

To address the shortcomings of existing research, the specific aims of the study were to:

- Evaluate the sustainability of inductees' trajectories of resilient functioning across three time points of measurement pre, post, and three months, following a five-day OA programme (increase/maintenance of impact/return to homeostasis) by total population, academic cohort, and gender;
- Determine the links of sustainable profiles of resilience to prospective benchmarks of achievement in HE (end of Year 1 academic outcomes) by total population and gender.

## 2. Method

### 2.1. Participants

The purposive sample included 2659 first year inductees, recruited over five consecutive years from 16 full-time sports courses in a single UK university. Females numbered 1279 (48.10%); the mean (M) age of the sample was 18.70 ($\pm$SD1.66), ranging 18 to 49 years. In terms of ethnicity, 96.8% were White, 0.6% Black, 0.6% Asian, 0.2% Chinese, and 1.9% were 'Other', making this sample highly compatible with a normative UK HE population of inductees. In screening for incomplete questionnaires or unmatched responses, the number of participants varied across the three time points of measurement. At pre-programme (Time 1, T1), 2659 participants were measured immediately on arrival at the residential centre. On the last day (post-programme, Time 2, T2), there were 2547 respondents. Three months later, at follow-up (Time 3, T3), data were captured from 1051 students. From T1 to T2, this represented a decline of 4.21%, rising to 60.47% at T3. Data relating to prospective academic achievement were limited to the first four annual cohorts and successfully matched to 1515 (59.48%) inductees.

### 2.2. Design

The principal design of the study was three repeated measures of the psychological resilience of HE inductees, prior to and following involvement in a five-day OA residential programme. The programmes were delivered during the first month of students' arrival at university and involved taking personal risks (perceived, rather than real—real risks were minimised) through progressive exposure to a range of activities designed to develop their adaptive capabilities. These activities included team challenges, educational visits, rock climbing and abseiling, ghyll scrambling, bivouacking, mountain-walking, canoeing, and kayaking, supported by post-event debriefing sessions. Sustained changes in resilience were also linked to prospective benchmarks of academic achievement and single response questions, regarding students' perceived well-being.

Despite not having a formal comparison condition across all time points, significant participant involvement ensured the power of the study was high for detecting change across each of the three time points. Procedural controls were provided by the stability of responses at T1 and from T2 to T3. Lack of differences between T2 and T3 may have suggested that any effects between T1 and T2 were due to the OA residential. Indeed, the significant differences in resilience between T1 and T2 had already been attributed to specific camp-based experiences of students who had attended the OA programme, compared to non-attendees [30]. For enduring impact, T2 and T3 resilience scores should have been similar. Any continued improvement suggested progressive and/or delayed effects, whereas decreased scores indicated shifts towards homeostasis (pre-OA residential levels).

Following institutional ethical approval, this study comprised of two stages of investigation, which matched the specific research purposes of the study. All analyses were conducted using the Statistics Package for the Social Sciences (SPSS), version 27 [77].

*2.3. Measures and Analyses*

2.3.1. Stage 1. Longitudinal Impact of OA on Inductees' Resilience

The Connor–Davidson Resilience Scale (CD-RISC) [78] was used to measure psychological resilience. This scale is suitable for use with older adolescents in educational contexts [79] and within OA residential interventions (e.g., [30]). Based on 25 phrases, CD-RISC generates total resilience (TR) (0–100) and five contributory subscale scores of (i) competence (0–32) (high standards and tenacity), (ii) trust (0–28) (trust in one's instincts, tolerance of stress), (iii) change (0–20) (acceptance, secure relationships), (iv) control (0–12) (self-determination in acquiring goals), and (v) spirituality (0–8) (faith in God or fate). CD-RISC items, e.g., 'I adapted to change', were scored 0 to 4, where 0 = not at all true and 4 = true nearly all the time; higher scores reflect greater resilience.

In university students, normative values of TR ranged from 55.8 [80] to 77.8 [81]. TR internal consistency (Cronbach's $\alpha$) was 0.92, and the subscales were as follows: competence (0.86), trust (0.77), change (0.75), control (0.65), and spirituality (0.60). Test–retest reliability demonstrated a high level of agreement with an intra-class correlation coefficient of 0.87. A construct validity was confirmed within the original validation of the scale [78].

Descriptive analyses of total resilience (TR) and subscale scores for each time point was undertaken for all inductees by gender and annual cohort. Repeated measures analyses of variance (ANOVA) established the significance of differences in TR (TR diff) and five subscales across three time points for all inductees, by gender and annual cohort. Paired *t* tests evaluated these differences between specific time points (T2 to T3 and T1 to T3). Cohen's *d* effect sizes (ES) substantiated statistical significance, highlighting the magnitude and direction of change in the mean outcomes of TR and subscales between time points. By convention, an ES of approximately 0.20 is 'small', ~0.50 'moderate', and 0.80+ 'high' [82].

Independent *t* tests and a one-way analysis of variance (ANOVA) established the significance of differences in TR diff and subscales between time points for gender and annual cohorts. Multivariate analysis of variance (MANOVA) evaluated the interaction of gender and annual cohort for TR diff and subscales between time points.

Every effort was made to minimise the number of missing respondents on the CD-RISC instrument across the three time points. Losses were negligible at T1 and T2, however the complexity of data collection across many courses and non-attendance of students provided substantial losses at T3. To cope with such interrupted data collection, one widely used, single imputation method is last observation carried forward (LOCF). This analysis attributes the last measured value of the endpoint to all subsequent, scheduled (but missing) evaluations. For a study with outcomes measured at multiple time points (repeated measures), if the endpoint analysis is used as an efficacy variable, the change from baseline to the last measurement is the dependent variable.

LOCF may produce biased estimates of interventions. For example, regardless of actual effects, the assumption of no change risks over- or under-estimating impact [83]. To minimise this potential problem, a conservative assumption of 'no change' was made, meaning missing data at T3 were replaced by respondents', corresponding to pre-programme (T1) scores.

Therefore, repeated measures analysis of CD-RISC operated with two separate matched data sets across the three time points. Titled the Missing Data Set, this included T3 data containing CD-RISC losses matched with respondents' corresponding T1 and T2 scores (*n* = 982). The LOCF data set included T3 data, supplemented by missing respondents' corresponding T1 scores (*n* = 2541). Both data sets were subject to the same statistical data analyses at this stage of the investigation.

2.3.2. Stage 2. Personal Well-Being Questions at Follow-Up

To explore the longevity of the impact of the OA programme, various aspects of inductees' subjective well-being, aligned to their resilience, was measured through a number of single response questions, three months following the OA programme. Current levels of optimism, feelings of control, and commitment to life as a student were established

through responses ranging from 1 (not at all) to 5 (significantly). Inductees were asked to report the frequency of their recollections of residential experiences, ranging from 1 (never) to 5 (through most days). The extent to which these experiences enabled students to settle into university life was investigated through the responses ranging from 1 (not at all) to 5 (significantly). To establish if any major incident may have influenced these follow-up measures at T3, inductees were requested to signify if any significant life event (1 (not at all), 2 (yes, positive event), and 3 (yes, negative event)) had occurred between the end of the OA programme (T2) to T3. Analyses of responses to all of these measures were descriptively portrayed.

2.3.3. Stage 3. Resilience Profiles and Prospective Academic Attainment

Bivariate correlations established the degree of association between TR and subscales to end of Year 1 academic outcomes. HE classifications included failure to complete studies/withdrawals and four separate grade boundaries, ranging from third (lowest) to first, which is the highest. Binary logistic regressions tested the sensitivity of inductees' resilience for predicting membership of groups of prospective academic attainment.

Tertiles of CD-RISC TR at T3 (highest, mid, and lowest third of scores) were calculated to explain the probability of individuals falling into one of two categories of end of Year 1 academic attainment. The validity of outcomes was evaluated using tests of the significance of predictions, levels of variance, and goodness of fit of models. To establish the relationship of TR differences between specific time points (T1 to T3 and T2 to T3) to prospective academic achievement, tertile groups for mean TR diff were subjected to the same binary logistic regression analyses. Both LOCF and Missing Data Sets were evaluated at this stage of the investigation.

**3. Results**

*3.1. Stage 1: Longitudinal Impact of OA Programmes on Inductees' Resilience*

Table 1 outlines mean pre-programme (T1), post-programme (T2), and 3-month follow-up scores (T3) for total resilience (TR) and subscales for all cohorts by gender. Matched data are included at T3 for the Missing Data Set (*n* = 982) and LOCF data set (*n* = 2541). Repeated measures ANOVA revealed significant mean differences in TR (TR diff) and all subscales between the three time points of measurement for all inductees (F values for TR and each subscale are presented). In both data sets, scores for TR and subscales increased in TR from T1 to T2, followed by decreases at T3. There were no significant differences in the TR and subscales at each time point between males and females, in either data set, with the exception of Spirit, where females scored higher. A gender x time point interaction was not significant for TR and all subscales for both data sets.

In both data sets (Figure 2) TR trajectories displayed a resilience 'witches hat' profile across the three time points. This highlighted increases at T2, returning to pre-programme levels (T1) three months later (T3). Reflecting an efficacious OA intervention, both data sets recorded T2 scores for TR that exceeded the normative value scores for undergraduate students' resilience that were reported in previous studies.

**Table 1.** Pre, post, and follow-up mean scores for total resilience (TR) and subscales by gender (Missing Data and last observation carried forward (LOCF) datasets).

| | Missing Data Set *n* = 982 (Females *n* = 507) | | | | | | | LOCF Data Set *n* = 2541 (Females *n* = 1234) | | | | | | |
| --- | --- | --- | --- | --- | --- | --- | --- | --- | --- | --- | --- | --- | --- | --- |
| | Pre (Time 1) | | Post (Time 2) | | Follow-Up (Time 3) | | | Pre (Time 1) | | Post (Time 2) | | Follow-Up (Time 3) | | |
| **Variables (Range)/Gender** | **M** | **±SD** | **M** | **±SD** | **M** | **±SD** | **F** | **M** | **±SD** | **M** | **±SD** | **M** | **±SD** | **F p < 0.001** |
| **TR (0–100)** | 72.03 | 12.28 | 78.15 | 11.77 | 71.63 | 12.51 | $F_{(2,1962)} = 171.55$ | 74.77 | 12.60 | 79.47 | 12.06 | 74.62 | 12.73 | $F_{(2,5080)} = 415.37$ |
| Male | 72.38 | 12.21 | 78.09 | 11.56 | 71.18 | 12.46 | $F_{(2,1960)} = 171.57$, $p < 0.001$ | 74.64 | 12.38 | 78.97 | 11.88 | 74.22 | 12.56 | $F_{(2,5078)} = 415.35$, $p < 0.001$ |
| Female | 71.71 | 12.36 | 78.20 | 11.98 | 72.05 | 12.54 | $F_{(2,1960)} = 171.57$, $p < 0.001$ | 74.90 | 12.84 | 79.99 | 12.22 | 75.05 | 12.89 | $F_{(2,5078)} = 415.35$, $p < 0.001$ |
| Male v Female | | | | | | | $F_{(1,980)} = 0.026$, $p = 0.872$ | | | | | | | $F_{(1,2539)} = 2.52$, $p = 0.113$ |
| Gender × Time | | | | | | | $F_{(2,1960)} = 1.90$, $p = 0.150$ | | | | | | | $F_{(2,5078)} = 2.132$, $p = 0.119$ |
| **Competence (0–32)** | 25.29 | 4.33 | 27.52 | 3.96 | 24.83 | 4.51 | $F_{(2,1962)} = 1.88.50$ | 26.06 | 4.40 | 27.79 | 4.06 | 25.88 | 4.50 | $F_{(2,5080)} = 416.02$ |
| Male | 25.50 | 4.45 | 27.74 | 3.89 | 24.87 | 4.50 | $F_{(2,1964)} = 188.85$, $p < 0.001$ | 26.10 | 4.37 | 27.79 | 4.03 | 25.87 | 4.44 | $F_{(2,5082)} = 415.68$, $p < 0.001$ |
| Female | 25.29 | 4.33 | 27.31 | 4.02 | 24.79 | 4.51 | $F_{(2,1964)} = 188.85$, $p < 0.001$ | 26.02 | 4.34 | 27.78 | 4.09 | 25.89 | 4.57 | $F_{(2,5082)} = 415.68$, $p < 0.001$ |
| Male v Female | | | | | | | $F_{(1,980)} = 2.08$, $p = 0.149$ | | | | | | | $F_{(1,2539)} = 0.021$, $p = 0.886$ |
| Gender × Time | | | | | | | $F_{(2,1960)} = 0.874$, $p = 0.417$ | | | | | | | $F_{(2,5078)} = 0.274$, $p = 0.760$ |
| **Trust (0–28)** | 19.05 | 3.89 | 20.49 | 4.12 | 19.00 | 4.08 | $F_{(2,1962)} = 76.77$ | 19.87 | 3.92 | 20.89 | 4.08 | 19.85 | 3.99 | $F_{(2,5080)} = 145.93$ |
| Male | 19.21 | 3.85 | 20.44 | 4.12 | 18.86 | 4.07 | $F_{(2,1966)} = 76.63$, $p < 0.001$ | 19.92 | 3.91 | 20.76 | 4.09 | 19.79 | 4.02 | $F_{(2,5082)} = 147.04$, $p < 0.001$ |
| Female | 18.90 | 3.92 | 20.54 | 4.12 | 19.13 | 4.07 | $F_{(2,1966)} = 76.63$, $p < 0.001$ | 19.81 | 3.93 | 21.03 | 4.07 | 19.91 | 3.98 | $F_{(2,5082)} = 147.04$, $p < 0.001$ |
| Male v Female | | | | | | | $F_{(1,980)} = 0.016$, $p = 0.898$ | | | | | | | $F_{(1,2539)} = 0.469$, $p = 0.493$ |
| Gender × Time | | | | | | | $F_{(2,1960)} = 2.33$, $p = 0.098$ | | | | | | | $F_{(2,5078)} = 3.68$, $p = 0.025$ |

**Table 1.** *Cont.*

| | Missing Data Set $n = 982$ (Females $n = 507$) | | | | | | | LOCF Data Set $n = 2541$ (Females $n = 1234$) | | | | | | |
| --- | --- | --- | --- | --- | --- | --- | --- | --- | --- | --- | --- | --- | --- | --- |
| | Pre (Time 1) | | Post (Time 2) | | Follow-Up (Time 3) | | | Pre (Time 1) | | Post (Time 2) | | Follow-Up (Time 3) | | |
| Variables (Range)/Gender | M | ±SD | M | ±SD | M | ±SD | F | M | ±SD | M | ±SD | M | ±SD | F $p < 0.001$ |
| **Change (0–20)** | 15.05 | 3.06 | 16.29 | 2.78 | 15.34 | 2.87 | $F(2,1962) = 75.73$ | 15.61 | 3.08 | 16.54 | 2.84 | 15.73 | 2.99 | $F(2,5080) = 188.45$ |
| Male | 15.13 | 2.96 | 16.19 | 2.80 | 15.17 | 2.86 | $F(2,1974) = 75.33$, $p < 0.001$ | 15.61 | 3.01 | 16.37 | 2.84 | 15.62 | 2.97 | $F(2,5090) = 190.21$, $p < 0.001$ |
| Female | 14.97 | 3.15 | 16.38 | 2.75 | 15.49 | 2.88 | $F(2,1974) = 75.33$, $p < 0.001$ | 15.62 | 3.15 | 16.72 | 2.82 | 15.84 | 3.00 | $F(2,5090) = 190.21$, $p < 0.001$ |
| Male v Female | | | | | | | $F(1,980) = 0.692$, $p = 0.406$ | | | | | | | $F(1,2539) = 3.63$, $p = 0.57$ |
| Gender × Time | | | | | | | $F(2,1960) = 2.82$, $p = 0.060$ | | | | | | | $F(2,5078) = 5.39$, $p = 0.006$ |
| **Control (0–12)** | 8.76 | 1.94 | 9.61 | 1.91 | 8.68 | 1.96 | $F(2,1962) = 117.00$ | 9.04 | 1.98 | 9.80 | 1.91 | 9.01 | 1.99 | $F(2,5080) = 345.31$ |
| Male | 8.83 | 1.92 | 9.73 | 1.85 | 8.67 | 1.96 | $F(2,1966) = 117.84$, $p < 0.001$ | 9.02 | 1.93 | 9.81 | 1.87 | 8.96 | 1.95 | $F(2,5082) = 343.83$, $p < 0.001$ |
| Female | 8.70 | 1.95 | 9.49 | 1.96 | 8.70 | 1.96 | $F(2,1966) = 117.84$, $p < 0.001$ | 9.06 | 2.02 | 9.79 | 1.95 | 9.07 | 2.02 | $F(2,5082) = 343.83$, $p < 0.001$ |
| Male v Female | | | | | | | $F(1,980) = 1.36$, $p = 0.243$ | | | | | | | $F(1,2539) = 0.456$, $p = 0.500$ |
| Gender × Time | | | | | | | $F(2,1960) = 2.05$, $p = 0.128$ | | | | | | | $F(2,5078) = 1.94$, $p = 0.144$ |
| **Spirit (0–8)** | 3.91 | 1.90 | 4.27 | 1.99 | 3.81 | 1.96 | $F(2,1962) = 31.01$ | 4.20 | 1.90 | 4.44 | 1.98 | 4.15 | 1.93 | $F(2,5080) = 53.18$ |
| Male | 3.73 | 1.93 | 4.00 | 2.00 | 3.63 | 2.01 | $F(2,1978) = 30.62$, $p < 0.001$ | 4.00 | 1.95 | 4.22 | 2.04 | 3.96 | 1.98 | $F(2,5092) = 53.12$, $p < 0.001$ |
| Female | 4.09 | 1.85 | 4.52 | 1.94 | 3.98 | 1.90 | $F(2,1978) = 30.62$, $p < 0.001$ | 4.20 | 1.90 | 4.67 | 1.89 | 4.36 | 1.85 | $F(2,5092) = 53.12$, $p < 0.001$ |
| Male v Female | | | | | | | $F(1,980) = 16.35$ | | | | | | | $F(1,2539) = 38.04$ |
| Gender × Time | | | | | | | $F(2,1960) = 1.207$, $p = 0.299$ | | | | | | | $F(2,5078) = 0.451$, $p = 0.637$ |

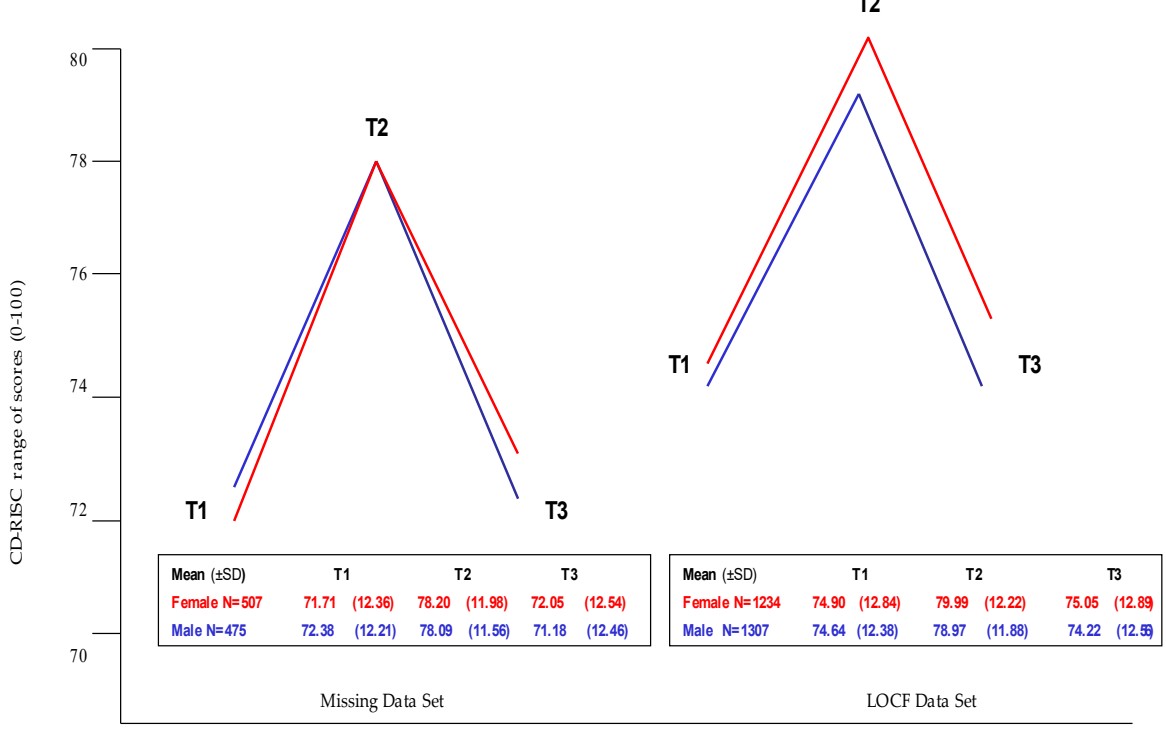

CD-RISC Total Resilience pre, post, follow-up

**Figure 2.** Pre (T1), post (T2) and follow-up (T3) mean Total Resilience by gender.

Table 2 provides TR and subscale mean scores for T1, T2, and T3, by annual cohort, for both data sets. Repeated measures ANOVA revealed that the mean TR (and subscales) for the three time points was significantly different within annual cohorts (Missing Data Set: $F_{(8, 1944)} = 32.42$, $p < 0.001$, and LOCF data set: $F_{(8, 5062)} = 31.92$, $p < 0.001$) and between annual cohorts (Missing Data Set: $F_{(4, 972)} = 79.30$, $p < 0.001$, and LOCF data set: $F_{(4, 2531)} = 333.77$, $p < 0.001$. Nonetheless, both data sets, across cohorts, portrayed similar patterns of immediate increases in TR from T1 to T2, followed by a subsequent decrease at T3.

To assess the specific longitudinal impact of the OA residential programme, two planned comparisons of TR and subscale scores were conducted for the Missing Data Set and LOCF data set. The first analysis compared the scores at T2 and T3 to ascertain the extent that resilience was increased, maintained, or decreased, three months following the OA programme. The second analysis identified differences between scores at T1 and T3, to highlight the sustainability of any changes (Table 3).

**Table 2.** Pre, post, and follow-up mean scores for total resilience (TR) by annual cohort for Missing Data and last observation carried forward (LOCF) data sets.

| Cohort / N/ variables (range) | Missing Data Set N = 982 | | | | | | LOCF Data Set N = 2541 | | | | | |
|---|---|---|---|---|---|---|---|---|---|---|---|---|
| | Pre (Time 1) | | Post (Time 2) | | Follow-Up (Time 3) | | Pre (Time 1) | | Post (Time 2) | | Follow-Up (Time 3) | |
| | M | ±SD | M | ±SD | M | ±SD | M | ±SD | M | ±SD | M | ±SD |
| **Year 1** | | | | | | | | | | | | |
| TR (0–100) | 68.04 | 8.94 | 70.36 | 10.94 | 67.66 | 10.17 | 67.93 | 9.41 | 68.17 | 10.48 | 67.81 | 9.81 |
| Competence (0–32) | 24.03 | 3.71 | 25.88 | 4.27 | 24.22 | 4.10 | 23.81 | 3.95 | 25.18 | 4.35 | 23.87 | 4.08 |
| Trust (0–28) | 18.12 | 3.12 | 17.91 | 3.36 | 17.35 | 3.41 | 18.36 | 3.16 | 17.84 | 350 | 18.11 | 3.34 |
| Change (0–20) | 14.01 | 2.05 | 14.56 | 2.59 | 14.37 | 2.71 | 13.93 | 2.37 | 14.25 | 2.54 | 14.05 | 2.59 |
| Control (0–12) | 8.24 | 1.66 | 8.50 | 1.89 | 8.27 | 1.81 | 8.15 | 1.72 | 8.42 | 1.81 | 8.16 | 1.76 |
| Spirit (0–8) | 3.63 | 1.83 | 3.48 | 2.07 | 3.39 | 1.71 | 3.67 | 1.89 | 3.48 | 2.00 | 3.59 | 1.86 |
| **Year 2** | | | | | | | | | | | | |
| TR | 74.82 | 11.94 | 76.67 | 10.68 | 71.05 | 12.23 | 74.57 | 11.83 | 76.46 | 10.85 | 72.55 | 12.09 |
| Competence (0–32) | 26.14 | 3.94 | 26.65 | 3.73 | 24.48 | 4.28 | 25.91 | 3.99 | 26.31 | 3.81 | 25.02 | 4.20 |
| Trust (0–28) | 20.16 | 3.95 | 20.53 | 4.03 | 19.12 | 4.20 | 20.30 | 3.86 | 20.66 | 3.96 | 19.74 | 4.06 |
| Change (0–20) | 15.84 | 2.63 | 15.88 | 2.68 | 15.31 | 2.89 | 15.70 | 2.67 | 15.85 | 2.66 | 15.41 | 2.81 |
| Control (0–12) | 8.97 | 1.91 | 9.44 | 1.83 | 8.67 | 1.93 | 8.89 | 1.91 | 9.37 | 1.88 | 8.73 | 1.92 |
| Spirit (0–8) | 3.74 | 1.91 | 4.20 | 1.96 | 3.49 | 2.03 | 3.79 | 1.84 | 4.29 | 1.93 | 3.66 | 1.92 |
| **Year 3** | | | | | | | | | | | | |
| TR | 77.84 | 14.12 | 88.13 | 6.54 | 79.49 | 12.90 | 81.87 | 11.73 | 88.76 | 6.22 | 82.54 | 10.98 |
| Competence (0–32) | 27.19 | 4.66 | 30.34 | 2.32 | 27.01 | 4.75 | 28.39 | 3.89 | 30.44 | 2.12 | 28.32 | 3.95 |
| Trust (0–28) | 20.45 | 4.12 | 23.38 | 2.75 | 21.28 | 3.92 | 21.57 | 3.62 | 23.59 | 2.74 | 21.90 | 3.46 |
| Change (0–20) | 15.85 | 4.05 | 18.37 | 1.65 | 16.93 | 2.72 | 16.95 | 3.37 | 18.45 | 1.75 | 17.37 | 2.69 |
| Control (0–12) | 9.67 | 1.93 | 10.81 | 1.42 | 9.58 | 1.94 | 10.05 | 1.75 | 10.95 | 1.28 | 10.03 | 1.76 |
| Spirit (0–8) | 4.75 | 1.87 | 5.25 | 1.68 | 4.74 | 1.91 | 4.92 | 1.78 | 5.33 | 1.69 | 4.91 | 1.80 |
| **Year 4** | | | | | | | | | | | | |
| TR | 64.30 | 9.29 | 84.28 | 11.80 | 67.84 | 11.50 | 78.99 | 12.74 | 87.52 | 8.11 | 79.67 | 12.38 |
| Competence (0–32) | 22.93 | 4.03 | 29.25 | 3.96 | 23.37 | 4.41 | 27.38 | 4.32 | 30.28 | 2.62 | 27.46 | 4.31 |
| Trust (0–28) | 16.23 | 2.64 | 22.25 | 3.98 | 18.22 | 3.84 | 20.67 | 4.02 | 22.90 | 3.09 | 21.04 | 3.86 |
| Change (0–20) | 13.26 | 2.85 | 17.68 | 2.52 | 14.51 | 2.74 | 16.49 | 3.13 | 18.35 | 1.92 | 16.73 | 2.89 |
| Control (0–12) | 7.93 | 1.93 | 10.48 | 1.81 | 8.22 | 2.04 | 9.57 | 2.03 | 10.90 | 1.40 | 9.63 | 2.01 |
| Spirit (0–8) | 3.95 | 1.65 | 4.62 | 1.71 | 3.52 | 1.84 | 4.91 | 1.67 | 5.10 | 1.64 | 4.83 | 1.76 |
| **Year 5** | | | | | | | | | | | | |
| TR | 66.67 | 8.24 | 71.25 | 8.75 | 67.77 | 9.98 | 66.62 | 8.85 | 71.54 | 9.17 | 67.13 | 9.67 |
| Competence (0–32) | 25.29 | 4.33 | 27.52 | 3.96 | 23.91 | 4.04 | 23.61 | 3.79 | 25.90 | 3.77 | 23.79 | 3.96 |
| Trust (0–28) | 19.05 | 3.88 | 17.86 | 3.35 | 17.57 | 3.19 | 17.27 | 3.12 | 18.05 | 3.37 | 17.35 | 3.27 |
| Change (0–20) | 14.30 | 2.17 | 15.02 | 2.41 | 14.51 | 2.34 | 14.25 | 2.35 | 15.03 | 2.40 | 14.35 | 2.43 |

**Table 2.** *Cont.*

| | Missing Data Set N = 982 | | | | | | LOCF Data Set N = 2541 | | | | | |
| | Pre (Time 1) | | Post (Time 2) | | Follow-Up (Time 3) | | Pre (Time 1) | | Post (Time 2) | | Follow-Up (Time 3) | |
|---|---|---|---|---|---|---|---|---|---|---|---|---|
| Control (0–12) | 8.08 | 1.66 | 8.83 | 1.66 | 8.13 | 1.71 | 8.06 | 1.63 | 8.93 | 1.65 | 8.09 | 1.65 |
| Spirit (0–8) | 3.38 | 1.74 | 3.56 | 1.88 | 3.67 | 1.77 | 3.44 | 1.74 | 3.58 | 1.95 | 3.57 | 1.76 |

**Table 3.** Mean Total Resilience (TR) subscale paired mean differences, effect sizes and percentages between specific time-points for both data sets.

| Variable (Range) | Missing Data Set N = 982 | | | LOCF Data Set N = 2541 | | | Differences ($p < 0.01$) | | | | Cohen's d Effect Size (ES) (+/−) | | | | % Difference (+/−) | | | |
| | Pre (Time 1) | Post (Time 2) | FUp (Time 3) | Pre (Time 1) | Post (Time 2) | FUp (Time 3) | Miss Data T2–T3 | Miss Data T1–T3 | LOCF Data T2–T3 | LOCF Data T1–T3 | Miss Data T2–T3 | Miss Data T1–T3 | LOCF Data T2–T3 | LOCF Data T1–T3 | Miss Data T2–T3 | Miss Data T1–T3 | LOCF Data T2–T3 | LOCF Data T1–T3 |
|---|---|---|---|---|---|---|---|---|---|---|---|---|---|---|---|---|---|---|
| TR (0–100) | 72.03 (12.28) | 78.15 (11.77) | 71.63 (12.51) | 74.77 (12.60) | 79.47 (12.06) | 74.62 (12.73) | $t(981) = 15.48$ | NS | $t(2540) = 23.64$ | NS | −0.53 | −0.03 | −0.39 | −0.01 | −8.34 | −0.54 | −6.10 | −0.20 |
| Competence (0–32) | 25.29 (4.33) | 27.52 (3.96) | 24.83 (4.51) | 26.06 (4.40) | 27.79 (4.06) | 25.88 (4.50) | $t(981) = 17.10$ | NS | $t(2540) = 23.10$ | NS | −0.60 | −0.10 | −0.44 | −0.04 | −9.77 | −1.82 | −6.87 | −0.69 |
| Trust (0–28) | 19.05 (3.89) | 20.49 (4.12) | 19.00 (4.08) | 19.87 (3.92) | 20.89 (4.08) | 19.85 (3.99) | $t(981) = 10.31$ | NS | $t(2540) = 13.18$ | NS | −0.36 | −0.01 | −0.25 | −0.00 | −7.27 | −0.26 | −5.23 | −0.10 |
| Change (0–20) | 15.05 (3.06) | 16.29 (2.78) | 15.34 (2.87) | 15.61 (3.08) | 16.54 (2.84) | 15.73 (2.99) | $t(981) = 9.24$ | NS | $t(2540) = 14.51$ | NS | −0.34 | 0.09 | −0.28 | 0.03 | −5.83 | 1.93 | −4.90 | 0.77 |
| Control (0–12) | 8.26 (1.94) | 9.61 (1.91) | 8.68 (1.96) | 9.04 (1.98) | 9.80 (1.91) | 9.01 (1.99) | $t(981) = 12.98$ | NS | $t(2540) = 20.44$ | NS | −0.48 | 0.11 | −0.43 | −0.01 | −9.67 | 5.08 | −8.06 | −0.33 |
| Spirit (0–8) | 3.91 (1.90) | 4.27 (1.99) | 3.81 (1.96) | 4.20 (1.90) | 4.44 (1.98) | 4.15 (1.93) | $t(981) = 7.03$ | NS | $t(2540) = 8.42$ | NS | −0.23 | −0.05 | −0.15 | −0.03 | −10.77 | −2.55 | −6.53 | −1.19 |

NS: not significant, effect size (ES) 0.2—small, 0.5—moderate, 0.8—large (Cohen, 1988).

In the Missing Data Set, there was a significant difference t (981) = 15.48, *p* < 0.001 between TR at T2 (M = 78.15, ±SD = 11.77) and T3 (M = 71.63, ±SD = 12.51), representing a percentage decrease of 8.34% ('moderate' negative ES difference of −0.53). In the LOCF data set, there was also a significant difference t (2540) = 23.64, *p* < 0.001 between TR at T2 (M = 79.47, ±SD = 12.06) and T3 (M = 74.62, ±SD = 12.73), signifying a decrease of 6.10% ('small' to 'moderate' negative ES difference of −0.39). Paired *t* tests for all subscales revealed significant differences and negative ES changes (ranging from 'small' to 'moderate') between mean scores at T2 and T3. For both data sets there was no significant difference between TR and subscales at T1 and at T3, suggesting an overall return to pre-OA programme functioning.

For both data sets, independent *t* tests reported no significant mean differences by gender in TR diff and subscales between T2 and T3. This suggested males and females experienced similar decreases in resilience from post programme to follow-up. There were significant differences by gender in TR diff and specific subscales between T1 and T3. In the Missing Data Set, males reported small negative mean differences in TR diff (M = −1.25, (±SD = 12.89)) and trust (M = −0.33, (±SD = 4.28)), as well as a small positive difference in change (M = 0.07, (±SD = 3.51)). In contrast, females possessed small positive mean differences in TR (M = 0.35, (±SD = 11.83)), t (984) = 2.06, *p* < 0.050, trust (M = 0.23, (±SD = 4.11)) t (980) = 2.16, *p* < 0.050, and change (M = 0.49, (±SD = 3.26)) t (980) = *p* < 0.050. These significant differences at T3 were also replicated in the LOCF data set. These findings suggest that females' resilience decreased in line with males' resilience (T2 to T3); however, they retained more of their post programme increases (i.e., it was more enduring than males after three months).

One-way ANOVAs revealed significant differences in TR diff and all subscales between T2 to T3 and T1 to T3 by annual cohort for both data sets. Figure 3 illustrates TR trajectories for each annual cohort across all time points for the LOCF data set. This shows that the last three annual cohorts reported small improvements in TR at T3, compared to pre-programme levels. MANOVA highlighted that there was no interaction effect involving gender and annual cohort for TR diff and subscales between T2 and T3 for both data sets. This suggested consistency in the decrease of resilience for males and females across cohorts at follow-up. However, a similar analysis of variables between T1 and T3 indicated that females achieved more sustained resilience than males across both data sets.

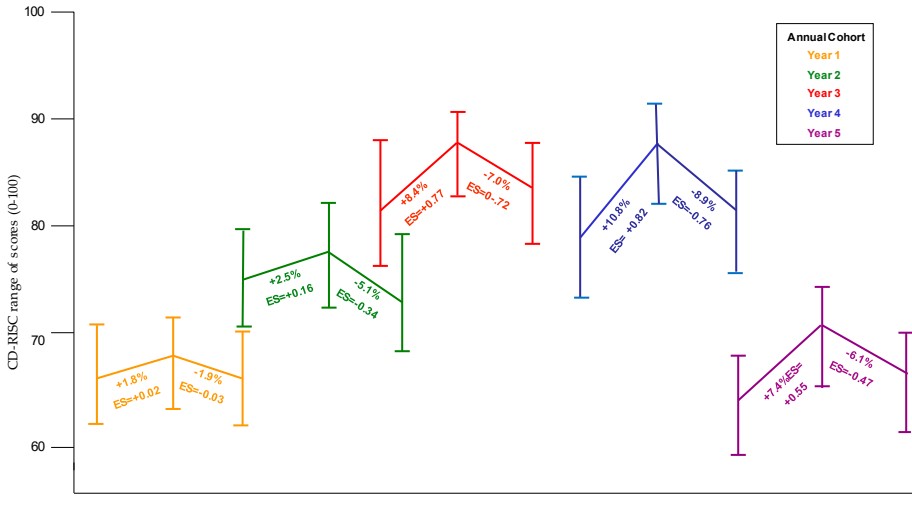

ES=Effect size 0.2 - small, 0.5 - moderate, 0.8 – large

**Figure 3.** Pre (T1), post (T2) and follow-up (T3) mean Total Resilience by annual cohort (LOCF Data Set).

*3.2. Stage 2. Personal Well-Being Questions at Follow-Up*

Inductees' personal wellbeing and recollections of OA residential experiences, three months following the programme, were represented through a number of single response questions. Figure 4 illustrates that four out of every five inductees had frequent recollections (ranging from '2 to 3 times' to 'through most days') of the OA programme. Less than 2 in 100 students had never thought about their OA residential experiences since the completion of the programme. This pattern of responses was evident for males and females.

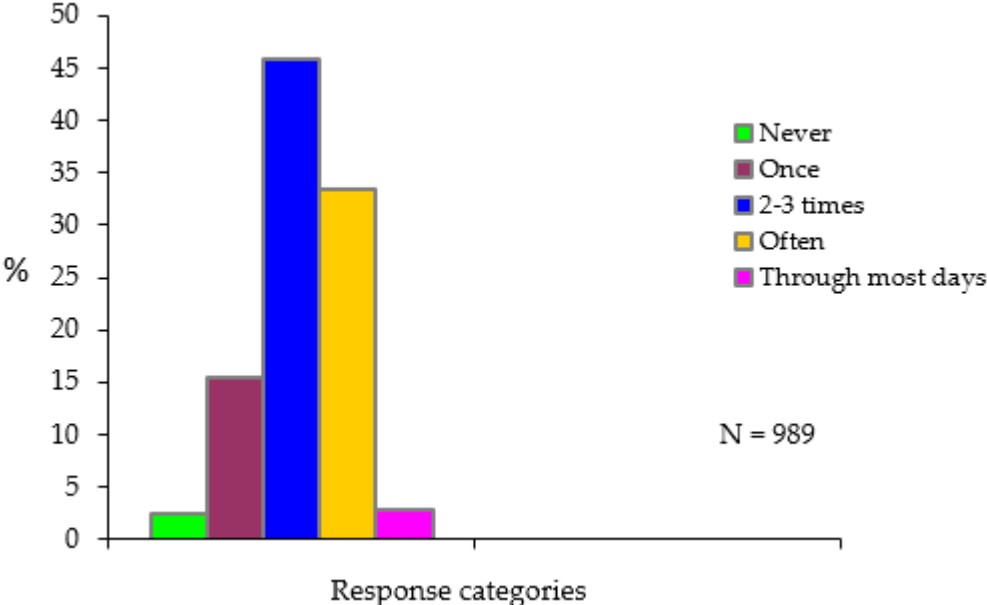

**Figure 4.** Thought of experiences' frequencies for all inductees.

OA residential programmes helped ease the transition of nine out of ten students into higher education (Figure 5). Only 2.2% of students perceived that the programme had not helped them settle into university life. Feelings of positive transition were reported more frequently by male inductees (11%). More males reported in the 'quite a lot' and 'significant' categories than their female counterparts.

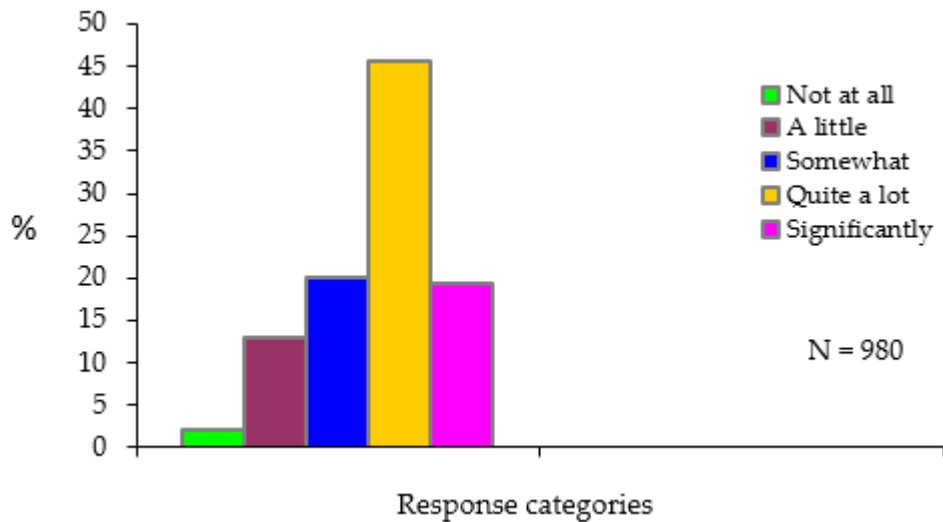

**Figure 5.** 'Help settle into life at university' frequencies.

Inductees were aware that questions regarding their current perceived level of optimism, personal control, and commitment to studies were conducted in the context of

evaluating any continued impact of the OA residential programme. Around 90% of students possessed optimism and felt in control of their lives at university three months following the programme. Males were inclined to report fewer negative responses and more positive answers (Figures 6 and 7).

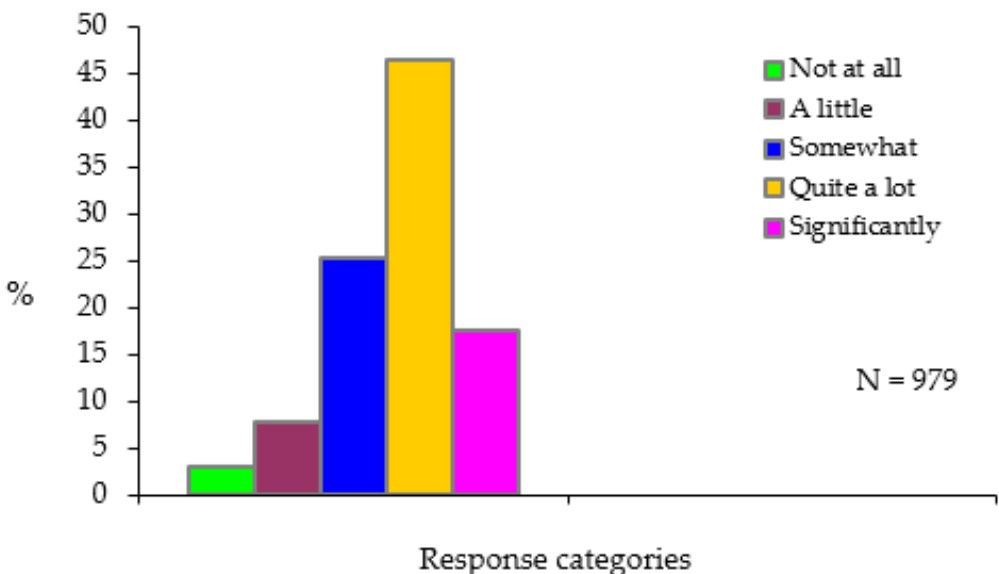

**Figure 6.** 'Optimisic you feel' frequencies.

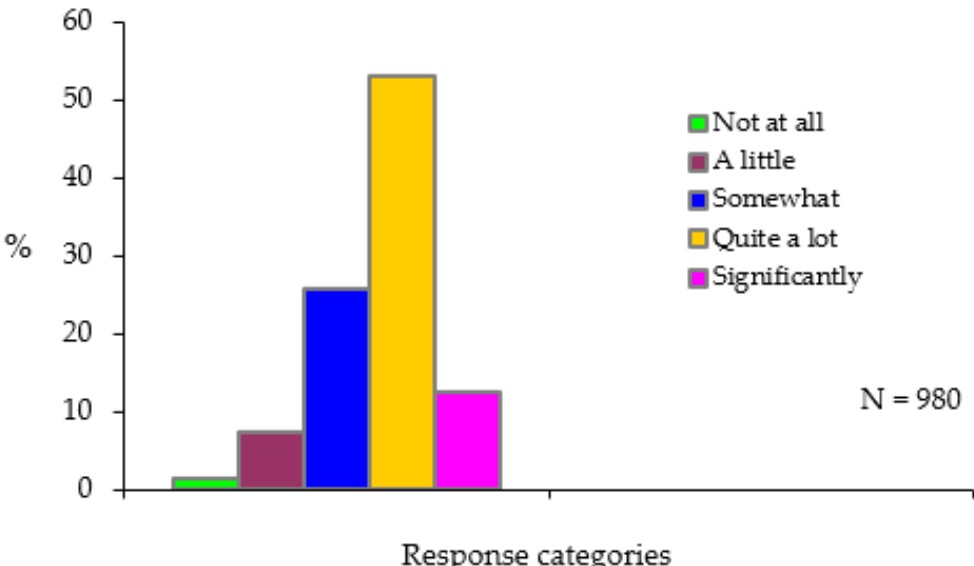

**Figure 7.** 'Control of life' frequencies.

Figure 8 highlights three-quarters of inductees were strongly committed to their life as a student. Less than 5% of students reported little or no commitment. Males frequented the higher two categories of positive responses (5% more) than females.

Figure 9 suggests that two-thirds of inductees did not experience a significant life event between the end of the OA residential and three months later. Over a quarter of students reported a positive event that was perceived as influential during this time. Females had 6% more noteworthy experiences in this period than males; mostly, these experiences were perceived as negative.

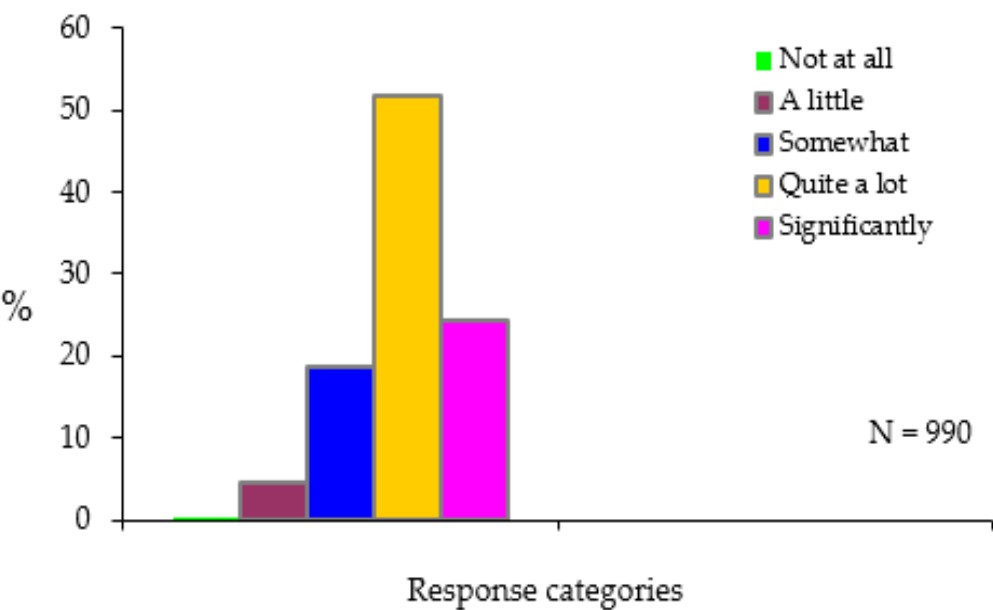

**Figure 8.** 'Committed to studies' frequencies.

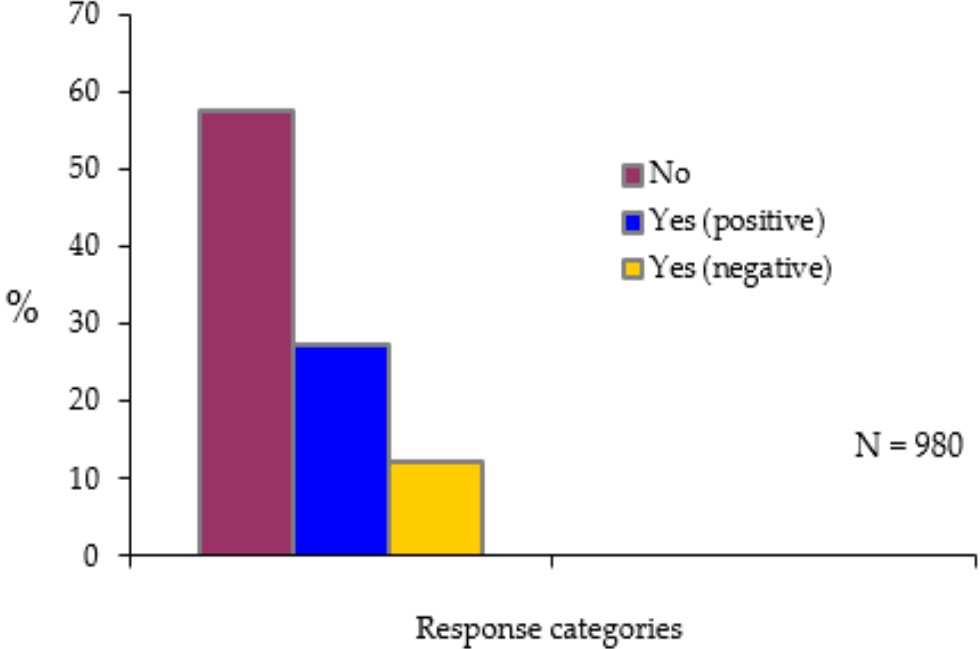

**Figure 9.** 'Significant life event' frequencies.

*3.3. Stage 3: Resilience Profiles and Prospective Academic Achievement*

Table 4 details inductees' mean follow-up (T3) TR subscales, corresponding to the end of Year 1 grade outcomes for both data sets. There were no significant differences in follow-up TR and subscales by gender with the exception of Spirit where females scored higher. Although females recorded a higher mean end of Year 1 grade, both gender groups acquired a 2.2 classification profile.

To provide more clarity to inductees' prospective academic achievement, Table 5 highlights the grade classification profile of end of Year 1 achievement. This highlighted that female inductees outperformed their male counterparts in the higher degree classifications and recorded 3% fewer third-class grades and fail/withdrawal.

**Table 4.** CD-RISC follow-up (T3) Total Resilience (TR) and subscales with mean end of Year 1 grades.

| | Mean (±SD) | | | | | |
|---|---|---|---|---|---|---|
| | **Missing Data Set** | | | **LOCF Data Set** | | |
| Variable (low to high range) | Males | Females | All | Males | Females | All |
| Sample *N* | *296* | *304* | *600* | *796* | *710* | *1506* |
| End of Year 1 grade + | 51.10 (13.12) | 53.72 (13.75) | 52.62 (13.53) | 50.77 (13.06) | 54.99 (13.86) | 52.76 (13.61) |
| CD-RISC Resilience subscales | | | | | | |
| Competence (0–32) | 24.85 (4.50) | 24.82 (4.47) | 24.83 (4.48) | 25.85 (4.45) | 25.89 (4.56) | 25.87 (4.51) |
| Trust (0–28) | 18.84 (4.06) | 19.14 (4.06) | 18.99 (4.06) | 19.78 (4.00) | 19.90 (3.99) | 19.84 (3.99) |
| Change (0–20) | 15.17 (2.85) | 15.47 (2.88) | 15.32 (2.87) | 15.62 (2.97) | 15.82 (3.01) | 15.71 (2.99) |
| Control (0–12) | 8.64 (1.99) | 8.69 (1.95) | 8.67 (1.97) | 8.93 (1.97) | 9.06 (2.02) | 8.99 (1.99) |
| Spirit (0–8) | 3.61 (2.01) | 4.00 (1.89) * | 3.81 (1.96) | 3.96 (1.98) | 4.38 (1.84) ** | 4.17 (1.93) |
| CD-RISC Total (0–100) | 71.09 (12.37) | 72.08 (12.48) | 71.59 (12.43) | 74.14 (12.55) | 75.04 (12.88) | 74.57 (12.72) |

**+** Grade classifications (40–50 Third, 50–60 2:2, 60–70 2:1, 70+ First),* Males v Females, indep *t* test, t (1020) = 3.19, *p* < 0.01, ** Males v Females, indep *t* test, t (2661) = 5.64, *p* < 0.01.

**Table 5.** End of Year 1 grade classifications (Cohorts 1–4) for all inductees and by gender.

| | Inductees | | | | | |
|---|---|---|---|---|---|---|
| | **Males** | | **Females** | | **All** | |
| End of Year 1 Outcome (grade range) | N | % | N | % | N | % |
| First (70+) | 6 | 0.8 | 23 | 3.2 | 29 | 1.9 |
| Upper Second (60–70) | 138 | 17.3 | 260 | 36.4 | 398 | 26.3 |
| Lower Second (50–60) | 413 | 51.6 | 329 | 46.0 | 742 | 49.0 |
| Third (40–50) | 169 | 21.1 | 55 | 7.7 | 224 | 14.8 |
| Fail/Withdrawal | 74 | 9.3 | 48 | 6.7 | 122 | 8.1 |
| Total | 800 | 100 | 715 | 100 | 1515 | 100 |

### 3.3.1. Follow-Up Total Resilience (TR)

Bivariate correlation analyses revealed positive associations between mean TR at T3 and end of Year 1 grade outcomes for all inductees in the Missing Data Set r (598) = 0.113, *p* < 0.01, and LOCF data set r (1504) = 0.089, *p* < 0.01. Small positive correlations were also observed in both data sets between inductees' end of Year 1 outcomes and all resilience subscales. Females recorded more positive correlations than males.

After confirming the appropriate tests for binary logistic regression (i.e., sample size, multi-collinearity, and goodness of fit), tertile groups of follow-up TR ('High' (82–100), 'Mid' (68–81), and 'Low' (29–67)) and gender were used to predict the probability of inductees falling into one of two categories of end of Year 1 outcomes. The interaction between tertile groups and gender for predicting academic outcomes was also investigated. Academic outcomes included comparisons of achieving the highest two grade classifications (First or 2.1) to all other grade outcomes, first and 2.1 outcomes, compared to 2.2. classification, 2.2 classification to lowest grade outcomes (third class/withdrawal/failure), and all grades to withdrawal/failure.

### 3.3.2. First and 2.1 Outcomes Compared to All Lower Grade Classifications

Analyses of both data sets found that inductees from the 'High' and 'Mid' tertile groups of follow-up TR were no more likely than students from the 'Low' tertile group to acquire the highest two grade classifications, compared to all lower grade boundaries. In the LOCF data set, 'High' tertile TR inductees (OR = 1.509, 95% C.I. 1.136 to 2.004)

were 1.5 times more likely than those in the 'Mid' group to achieve a first or 2.1 than a lower grade outcome ($\chi$2 (1, 1083) $p < 0.01$). Gender analyses demonstrated that female inductees were three times more likely than males to acquire a first or 2.1 than lower grade classifications in both data sets.

### 3.3.3. First and 2.1 Outcomes Compared to 2.2. Classification

Within both data sets, there were no differences in the likelihood of inductees from the 'High' and 'Mid' tertile groups than students from the 'Low' tertile group to acquire the highest two grade classifications, compared to a 2.2 classification. In the LOCF data set 'High' tertile follow-up TR inductees (OR = 1.525, 95% C.I. 1.136 to 2.055) were 1.5 times more likely than those in the 'Mid' group to achieve a first or 2.1 ($\chi$2 (1, 1083), $p < 0.01$), compared to a 2.2 outcome. Females were more than twice as likely as males to acquire a first or 2.1 than a 2.2 across both data sets.

### 3.3.4. 2.2. Classification to Lowest Grade Outcomes

Within the Missing Data Set, 'High' follow-up, TR tertile inductees ($\chi$2 (1,397), $p < 0.01$) were twice as likely as 'Mid' (OR = 1.939, 95% C.I. 1.048 to 3.558) and 'Low' range students (OR = 2.337, 95% C.I. 1.274 to 4.272) to achieve a 2.2 than a lower academic outcome. Within the LOCF Data Set, 'Mid' female inductees were twice as likely as 'mid' males to acquire a 2.2, rather than less competent outcomes, compared to 'low' TR tertile students. Females in both data sets were twice as likely as males to acquire a 2.2.

### 3.3.5. All Grades to Withdrawal/Failure

Missing and LOCF data sets reported no differences in the likelihood of inductees, from either follow-up TR tertile group or gender failing or withdrawing from the programme, compared to all higher-grade boundaries.

### 3.3.6. Tertiles of Total Resilience Difference (TR diff)

Tertiles for TR difference were created between T2 and T3 and T1 to T3. These were subject to the same logistic regression analyses as follow-up T3 TR tertiles. This process aimed to ascertain predictive links of grouped differences in TR between these time points and the prospective academic achievement. From T2 to T3, neither tertile group significantly predicted achievement between the grade boundaries within both data sets. In contrast, within the T1 to T3 analyses, significant differences were evident in the LOCF data set between tertile groups for being predictive of inductees acquiring higher academic attainment. 'Positive difference' inductees were 1.5 times more likely (OR = 1.544, 95% C.I. 1.161 to 2.052) than those in the 'Small negative difference/No change' group to achieve a first and 2.1 than a lower grade outcome ($\chi$2 (1, 333) $p < 0.01$); they were 2.5 times more likely (OR = 2.499, 95% C.I. 1.679 to 3.719) than the 'Small negative difference /No-change' group ($\chi$2 (1, 289) $p < 0.01$) and twice as likely (OR = 1.989, 95% 1.222 to 3.214) as the 'High negative difference group' to acquire a 2.2 than a lower outcome ($\chi$2 (1, 277) $p < 0.01$).

## 4. Discussion

This large empirical study investigated the sustainability and functionality of university inductees' self-reported psychological resilience, following attendance at OA induction residential programmes. Conceptually, experience of OA may promote protective resistance against on-going transitional stressors for a wide range of students. Two comparable data sets were used; these confirmed two major findings, in response to the research aims.

First, inductees' psychological resilience, across all annual cohorts, represented a consistent and reasonably rapid return to homeostasis once displaced (pre-programme, T1, to 3-month follow-up, T3), suggesting 'bounce-back-ability' and a stable trajectory of healthy functioning [48–50]. This transient perturbation reflected a significant decrease in resilience from widespread initial gains already attributed to the OA programme [30], followed by a general realignment to pre-programme levels. Resilience decreases at T3

were similar for both genders; however, male resilience was more enduring and facilitative for prospective academic success.

Second, specific subdomains of resilience (e.g., perceived control and interpersonal capacity), associated with academic success (e.g., [54]), were evident in students identified as being highly resilient at follow-up (T3). Importantly, these higher scores carried through to higher end-of-year grades, suggesting the possibility of an incremental value of high and/or increased resilience.

Findings from the present study suggested efficacious OA programming may have contributed to a pathway of adaptable, productive functioning for new students. Students' feelings of well-being, underpinning resilient behaviours (optimism, control), were reported at T3, aligned with frequent recollections of OA experiences, which had helped support their transition. However, due to the lack of a comparison group throughout all time points, such effects cannot be attributed to the OA programme. Although inductees' mean follow-up (T3) resilience was lower than resilience reported in previous much smaller studies (e.g., [31,72,73]), three annual cohorts reported small improvements in their resilience at T3 compared to pre-programme levels. Although these increases were non-significant, resilience reported at T3 was likely to be facilitative of future academic performance. These findings provide the basis for future study to help justify the purposive adoption of OA residential programming for generating healthy sustainable adaptive capacity for incoming university students.

### 4.1. Trajectories of Inductees' Resilience
Sustainability of Resilience

Our evidence confirms that inductees' resilience was receptive to positive change across three time points of measurement. The variability of responses across cohorts also suggests the importance of elements of the immediate/on-going OA experience. The reduction in mean resilience, three months following the OA programming, warrants attention and justifies the criticism directed at OA programming, regarding longevity of impacts (e.g., [74]). It also demonstrates the difficulties some individuals have in sustaining resilience [34]. For example, students who self-reported resilience in two annual cohorts were worse at T3, compared to pre-programme levels.

Given the complexity of resilience and diversity of the courses represented in the study, it is naive to expect that the 40+ OA programmes reported here would generate equivalent and positive life experiences. More consistent resilience responses may have been secured if programming incorporated bespoke teaching practices and learning retrieval processes that transferred into, and were activated within, HE. This could include infusing OA activities with authentic problem-solving and critical incidents—by connecting to core subject matter and course expectations, which require metacognition. The transfer of positive responses from the OA context to the HE context may rely on making a deep, personal connection to the new setting fostered through a sufficiently relevant and meaningful experience [26,69].

Nevertheless, even without such tailor-made provisions, a range of investigations, afforded by two extensive, highly compatible data sets, revealed students realigned their behaviours toward a healthy trajectory of resilient functioning. These are encouraging findings; in many cases, the gains exceed those reported by previous similar studies (e.g., [31,32,71–73]). Furthermore, the increases in the CD-RISC subscales (e.g., trust in one's instincts (denoting tolerance to stress) and change (reflecting sociability)) align closely with developmental competencies realised within transition-oriented OA programmes (e.g., [26,69]). Indeed, the items of the CD-RISC sub-domains seem readily adaptable into teachable behaviours and approaches; many have proven worth for HE retention and achievement.

### 4.2. Resilience and Prospective Academic Outcomes

Confirming the relationship between resilience, positive educational adjustment, and academic attainment (e.g., [58,60]), higher follow-up resilience was associated with bet-

ter prospective academic outcomes. These links may be sufficient to prompt the use of OA-related strategies to build adaptive capacity across inductee populations. Despite these findings, evidence from the current study also cautions against any expectation that higher resilience pre-determines better academic achievement. Confirming the fluidity, individuality, and temporal quality of resilience [43,70], some students reporting lower follow-up resilience were no less likely to acquire future academic success than those with higher values. Given resilience may help students to develop resources for dealing with study pressures, it's relative (more than absolute) value and facilitative use across time, which may be best promoted through OA programming.

### 4.3. Gender

Females' resilience was more enduring and functional than that of males. This aligns with previous studies following OA programming [71]. Females also increased the chances of acquiring better end of Year 1 grade outcomes. The female gender has been identified as a predictor of academic achievement in first year university students; generally, females have outperformed males across performance benchmarks in UK HE [52]. Without gender-stereotyping, the more nuanced form of enduring resilience reported among females, characterised by spirituality and relationships, may promote a collective buffering of stress that complements the collaborative teaching methods and continuous assessment increasingly favoured in universities [17]. Given the less receptive nature of males' resilience to these OA experiences, HE practices may need to be revised to help males to better align their adaptive capabilities to the challenges encountered in HE.

### 4.4. Strengths, Limitations and Future Considerations

This large, unique study presents the strengths and caveats within the procedures and findings. Missing data, at three months following the OA programme, provided a challenge for developing valid longitudinal profiles. Although there are limitations to the last observation carried forward (LOCF) procedure [82], this method was able to deliver conservative, yet comparable findings to the missing data set across three time points. The CD-RISC scale was sensitive to the differences associated with better and worse outcomes and confirms its use as a valid and reliable psychometric tool for measuring undergraduate students' resilience (e.g., [77,78]).

Notwithstanding its impressive scale across multiple courses of study and time, the key limitation in this study was the lack of a comparison condition at follow-up. This made it difficult to substantiate the lasting impact of the OA programme, following an established, significant increase in students' resilience. The single-response questions at follow-up (aspects of students' well-being and perceived impact of the programme) shed light on the frequently recalled lasting influence of the OA residential, as an efficacious method for supporting the transition into HE. Nevertheless, reliance on self-report data, rather than direct demonstration of resilience or more nuanced evaluations of adaptive behaviours, present difficulties for interpretation. Qualitative testimonies from students, tutors, and OA practitioners may have provided a range of perspectives for better understanding these perceived longitudinal changes.

Although the findings are limited to a single university, the expansive data capture established important, predictive links between those students who reported higher respective resilient behaviours at follow-up and the likelihood of prospective academic achievement. However, end of Year 1 outcomes represented only one aspect of competent functioning within HE; for some inductees, seemingly modest outcomes may have reflected considerable resilience.

Many risk factors contribute to the mental ill-health and the low resilience of young people starting university. These include family relationships, peer pressure, exam pressures, over-protective parenting, and bureaucracy, which may cocoon young people from failure. The pandemic and its lockdowns will exacerbate these risk factors and present new impediments for the positive assimilation of students into HE, for the short term and

months to follow. During this exceptionally difficult time, many students will require resilience, drawing on social support, strength-based learning, religious/spiritual beliefs, and other coping strategies to maintain (or even increase) psychological wellbeing during and following the pandemic.

Leveraging the evidence from this study may expose the requirement for active practices that promote enduring resilient responses within HE inductees to help them consistently weather a landscape of stressors. Results seem to indicate that these week-long OA programmes provided an immediate impact which dissipated relatively quickly. Nonetheless, the resonance of the experiences was perceived by most students to support their adaptive behaviours during the first three months of university life. The complexity of resilience and span of responses to OA programming within our findings confirms that young people differ in how they interact with challenges over time. This uncertainty may require that universities develop flexible interventions to help students cope with changing circumstances. However, the healthy trajectories revealed by this study provide an impetus to create more generic and tailored interventions, at key transitional points across the student lifecycle, that enhance students' adaptive capacities. This resilience may range from students using a range of learned skill sets for surviving threats to well-being and others developing forward momentum and thriving.

Future studies should investigate the nature of post event resilience resulting from longer OA exposure (continuous measurement of longitudinal responses), comparing these effects to a range of influential factors for positive adaptability. Furthermore, to build effective and sustainable interventions, it may be necessary to differentiate the key environmental and psychosocial factors associated with lasting changes in participants' lives. Analyses of the interaction of these factors may help to identify components of OA programming and behaviours that maximise opportunities for transfer across settings.

**Author Contributions:** Writing–original draft preparation, J.F.A.; review and editing J.M. All authors have read and agreed to the published version of the manuscript.

**Funding:** This research received no external funding.

**Institutional Review Board Statement:** The study was approved by the Ethics Committee of Leeds Beckett University.

**Informed Consent Statement:** Informed consent was obtained from all subjects involved in the study.

**Data Availability Statement:** The data presented in this study are available on request from the corresponding author. The data are not publicly available due to ethical/privacy issues.

**Acknowledgments:** Carnegie Great Outdoors for access to inductees attending a five-day OA residential programme. Permission was granted by the authors of CD-RISC for its use in this study.

**Conflicts of Interest:** The authors declare no conflict of interest.

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
