# Peer review of "Trajectories of Resilience in University Inductees following Outdoor Adventure (OA) Residential Programmes"

_2673-5318, doi:10.3390/psychiatryint3010007_

Round 1

Reviewer 1 Report

This manuscript reports on an evaluation of the impact of an outdoor adventure program (OA) on self-perceived resilience among university students in a specialized sports course. The study was described as “complex” by the authors. Yet, to me, it didn’t seem especially complicated – it was a simple, three-time design, assessing impact of a specific intervention. Unfortunately, the chopping and dissection in the data between different sub-groups and separated time periods, along with a rather heavy writing style, made the manuscript extremely complex and hard to follow. My sense was that the results, overall, indicated little lasting benefits from this OA program, yet the authors appeared to want to indicate a positive benefit and got themselves into statistical contortions to do so. A complete overhaul of the manuscript and data analyses, providing a much more straight-forward and clear description of the methods and its results, alongside a simple, basic analysis of the outcomes, would very much help to create a manuscript with a clear take-home message.

The authors need to much more clearly and strongly emphasize the key limitation in this study – the lack of any form of comparison condition. This needs to be acknowledged right from the outset of the discussion and in turn, the point needs to be clearly acknowledged that any effects cannot necessarily be attributed to the OA program. Another limitation that really needs some coverage is that what is assessed in this study is “self-perceived resilience” rather than any actual demonstration of resilience to life stressors.

As noted, the data analysis seemed unnecessarily complicated. Dividing the sample into tertiles simply maximizes chance findings and relies on an arbitrary grouping. Comparing cohorts adds additional complexity with no theoretical basis. It would be far easier for readers if the authors simply analyzed their results with a simple, one-way ANOVA. Comparing between the sexes could be done if required – in which case a simple, 2x3 ANOVA reporting two main effects and a time by sex interaction would suffice. This would greatly reduce the length and complexity and provide a clear and simple message. If the authors wish to evaluate the impact of baseline levels of perceived resilience, then some form of modelling (such as HLM) would allow a single analysis to predict individual outcomes from baseline intercept, rather than arbitrarily dividing the sample into three. On an additional statistical point, LOCF is typically viewed as a very poor method of missing data imputation and an alternate method such as multiple imputation or maximum likelihood would be far better.

The authors also need a more balanced approach to describing their conclusions. At present it felt as though the authors were putting a positive spin on the data. As just one example, in the discussion, p 5, the authors claim that 40% of students reported positive gains in resilience at follow-up. Yet in Table 4, they report that positive gains from T1 to T3 occurred in only 16% of students, which is no different to the 17% showing a negative change. Further, in this table, positive change is defined by a difference of as little as 1 point on the scale, which is well within the bounds of random error. Similarly, the interpretation that vulnerable inductees benefitted most is based on extremely small positive gains in this tertile that might easily be the result of regression to the mean and ceiling effects.

Reviewer 2 Report

The authors of this paper present a longitudinal study that aimed to identify the association of resilience with future benchmarks of achievement in higher education (academic performance at the end of the first year of study) and to assess the sustainability of resilient performance of participants over time. The study, although done on a large sample, unfortunately has significant limitations. In my opinion, the main one is the absence of a control group, which means that we cannot conclude from the results presented, but only talk about co-occurrence. The second major limitation is that it does not take into account other factors that we know are related to human resilience. Personality traits (openness, extraversion and agreeableness), internal locus of control, self-control, self-efficacy, self-esteem and clearly optimism contribute to resilience. Also intellectual functioning, cognitive flexibility, social attachment, positive self-concepts, emotion regulation, positive emotions, spirituality, active coping, resilience, optimism, hope, resourcefulness, and adaptability are associated with resilience.1 Unfortunately, this study did not take into account other factors that might have influenced the presented results. Considering the above, despite the large study group, the presented results should be analysed with great caution. Regardless of this, the idea behind the study is interesting and the results give grounds for further research in this direction. 

I would like to draw the authors' attention to several factors, the correction of which, in my opinion, would positively influence the presented work:

1/ Line 35-39 "Even before the effects of COViD-19 and lockdowns emerged, there were enduring concerns about the psychological development of the current cohort of young people. Risks to their mental health and productive functioning may come in the form of unemployment, overreductive education, fast-changing technological advancements, high divorce rates, media in trusion and consumerism." The lack of confirmation in the quoted literature of this statement suggests that the literature should be supplemented - where did the authors derive this knowledge from.

2/ Table 1 - in the header appears "F p<.001" and in this column appear the function and p values for the data "male v female" also with a p value. I am also missing function values for "Male" and "Female" I suggest filling in the missing function and p values for each variable presented in the table. In the header remove "p<.001" additionally in the line Trust/Gender x Time appears "p=..098" - please remove one dot. 

3/ Figure 2 - in my opinion the data would be better presented in a bar chart with labels presenting M and SD. Connecting the points T1/T2/T3 may suggest to the reader the continuity of the measurement. 

4/ Table 3. - move the heading "Means (±SD)" to make it clear what it refers to or put it in the table caption. "Effect size (ES) 0.2 - small, 0.5 - moderate, 0.8 - large" in the table seems redundant as the table gives numerical data and their interpretation is another matter. The abbreviation "NS" from the table should be explained

5/ The authors use the sentence "Effect size (ES) 0.2 - small, 0.5 - moderate, 0.8 - large" without giving the source - it should be supplemented and indicated where exactly such an interpretation of the effect size came from.

6/ Figure 4 - unfortunately descriptions are illegible letters/digits overlap - I propose to widen distances between T1 - T2 - T3 and mark these points on the axis instead of on the diagram also in other diagrams)

7/Propose to expand the "limitation" section by at least including the limiting factors indicated at the beginning of the study. 

  1. Joseph S, Linley PA. Growth following adversity: theoretical perspective and implications for clinical practice. Clin Psychol Rev. 2006;26(8):1041–1053.

Round 2

Reviewer 1 Report

I have re-read this paper following my previous review and the changes made by the authors. The removal of the tertile analyses and the minor changes to wording and clarifications have made this a somewhat easier paper to follow. I commend the authors for these adjustments and for their acknowledgement of some of the core methodological limitations.

My primary remaining concern is that I am still feel as though the authors are trying to provide an overly positive (and unscientific) interpretation of the findings in the discussion. In fact this was so much the case, that I began to wonder whether the authors had a financial interest in the OA program – although I then noted that they report no CoI. In looking at the “witches hats” for the primary outcomes (and the reported effect sizes), it seems clear to me that the OA program led to moderate increases in resilience from baseline to post-intervention, which rapidly returned to baseline levels three months later. In other words, there was a brief, temporary benefit (which has already been reported in another paper) and no sustained benefits. However, a reader of this paper who jumped to the discussion section without looking carefully through the results, would have the impression that this program led to “positive, lasting change” (p. 8 discussion). It is possible that I have misunderstood. But in that case, there is still an issue of clarity – the authors need to far more clearly justify each of their claims from their results. Or if I have not misunderstood, then the authors need to provide a far more scientifically impartial interpretation. For example, conclusions that “OA programming may have contributed to an enduring pathway of adaptable, productive functioning” or that resilience showed “positive lasting change”; or that “inductees sustained and even improved adaptive function” – need to be supported by the relevant sections of data, especially given that changes in resilience from T1 to T3 had effect sizes that were all less than 0.1 and were not statistically significant. Similarly, the authors need to be careful about “cherry-picking” data to support their ideal conclusions. By this I am referring to the repeated comment that “In three of the five cohorts a proportion sustained their resilience gains”. My interpretation of Fig 3 is that the difference from T1 to T3 for these three cohorts is extremely small (less than .1) and very unlikely to be significant. But nonetheless, if the authors wish to interpret these tiny effects as meaningful, then they should point out that for two cohorts, self-reported resilience at T3 was worse than at T1 – perhaps suggesting that the OA program may be associated with significant risk.

My only other minor suggestion is that I struggled to understand exactly how the “missing data” dataset was constructed and perhaps an extra sentence or two explaining this would be useful. In fact, I missed a simple sentence that clearly indicated the amount of missing data at each time point – e.g. “x students completed data at T1; y students (x% of baseline) completed data at T2; z students (x% of baseline) completed data at T3”. An indication of whether these data were likely to be missing at random would also be useful.

Finally, I think there is a small typo on p 8, line 344 – “across the three time points by”

In short I commend the authors for a nice study and some excellent data collection. I think this paper could provide a nice contribution to the literature. However, I do think that it needs to provide a scientifically impartial interpretation of the results in order to provide this contribution. Based on my reading, the results seem to indicate that these types of programs have a brief, “feel good” impact, but that this effect dissipates relatively quickly, underpinning the difficulties of producing lasting increases in perceived resilience with such brief programs.

Round 3

Reviewer 1 Report

The authors have made a few concessions to my earlier concerns. The discussion is perhaps still a somewhat "positively oriented" one, but this is now within acceptable limits. I have no further suggestions.